



**Measurement report: Brown Carbon Aerosol in Polluted Urban Air of North China Plain:**
**Day-night Differences in the Chromophores and Optical Properties**
**Yuquan Gong**[1,2]**, Ru-Jin Huang**[1,2,3,4]**, Lu Yang**[1,2]**, Ting Wang**[1]**, Wei Yuan**[1,2]**, Wei Xu**[1]**,**
**Wenjuan Cao**[1]**, Yang Wang** [5,6]**, Yongjie Li**[7]
[1] State Key Laboratory of Loess and Quaternary Geology, CAS Center for Excellence in
Quaternary Science and Global Change, Institute of Earth Environment, Chinese Academy of
Sciences, 710061 Xi'an, China
[2] University of Chinese Academy of Sciences, Beijing 100049, China
[3] Institute of Global Environmental Change, Xi'an Jiaotong University, Xi'an 710049, China
[4] Laoshan Laboratory, Qingdao 266061, China
[5] School of Geographical Sciences, Hebei Normal University, Shijiazhuang, China
[6] State Key Joint Laboratory of Environmental Simulation and Pollution Control, Beijing, China
[7] Department of Civil and Environmental Engineering, Faculty of Science and Technology,
University of Macau, Taipa, Macau SAR 999078, China
Correspondence: E-mail: rujin.huang@ieecas.cn (R.-J.H)





**Abstract.** Brown carbon (BrC) aerosol is light-absorbing organic carbon that affects radiative
forcing and atmospheric photochemistry. The BrC chromophoric composition and its linkage
to optical properties at the molecular level, however, are still not well characterized. In this
study, we investigate the day-night differences in the chromophoric composition (38 species)
and optical properties of water-soluble and water-insoluble BrC fractions (WS-BrC and WIS-
BrC) in aerosol samples collected in Shijiazhuang, one of the most polluted cities in China. We
found that the light absorption contribution of WS-BrC to total BrC at 365 nm was higher during
the day (62 ± 8%) than during the night (47 ± 26%), which is in line with the difference in
chromophoric polarity between daytime (more polar nitrated aromatics) and nighttime (more
less-polar polycyclic aromatic hydrocarbons, PAHs). The high polarity and water solubility of
BrC in daytime suggests the enhanced contribution of secondary formation to BrC during the
day. There was a decrease of the mass absorption efficiency of BrC from nighttime to daytime
(2.88 ± 0.24 vs. 2.58 ± 0.14 for WS-BrC and 1.43 ± 0.83 vs. 1.02 ± 0.49 $m^2$ g $C^{-1}$ for WIS-BrC,
respectively). Large polycyclic aromatic hydrocarbons (PAHs) with 4–6-rings PAHs and
nitrophenols contributed to 76.7% of the total light absorption between 300–420 nm at night
time, while nitrocatechols and 2–3-ring oxygenated PAHs accounted for 52.6% of the total light
absorption at day. The total mass concentrations of the identified chromophores showed larger
day-night difference during the low-pollution period (day-to-night ratio of 4.3) than during the
high-pollution period (day-to-night ratio of 1.8). The large day-night difference in BrC
composition and absorption, therefore, should be considered when estimating the sources,
atmospheric processes and impacts of BrC.



## 1 Introduction


Light-absorbing organic carbon aerosols, also termed brown carbon (BrC) aerosol, are
ubiquitous in the atmosphere (Iinuma et al., 2010; Yuan et al., 2016; Huang et al., 2021).
Growing evidence has shown that BrC can reduce atmospheric visibility, affect atmospheric
photochemistry, and change regional and global radiation balance (Kirchstetter et al., 2004;
Laskin et al., 2015; Hammer et al., 2016). Besides, some components in BrC, such as polycyclic
aromatic hydrocarbons (PAHs) are highly toxic and carcinogenic, which can adversely impact
human health (Alcanzare, 2006; Zhang et al., 2009; Huang et al., 2014). The extent of these
effects is closely related to the optical properties and chemical composition of BrC, which are
still not well understood.
BrC is often classified into water-soluble (WS-BrC) and water-insoluble (WIS-BrC)
fractions because these two fractions are largely different in chemical composition and light
absorption. For example, abundant nitrophenols were detected in WS-BrC, while polycyclic
aromatic hydrocarbons (PAHs) were the main component of WIS-BrC (Huang et al., 2018;
Huang et al., 2020). The difference in BrC chemical composition is associated with the emission
sources. For example, methyl nitrocatechols are specific to biomass burning, while PAHs are
mainly emitted by fossil fuel combustion (Kitanovski et al., 2012; Dat and Chang, 2017).
Atmospheric oxidation can further complicate the BrC chromophores dynamically, leading to
light-absorbing enhancement or bleaching. For example, Li et al. (2020) reported that the mass
absorption efficient (MAE) of some nitroaromatic compounds (e.g., nitrocatechols) from the
biomass burning can enhance about 2−3 times by oxidation to generate secondary
chromophores. Yet, prolonged photo-oxidation reactions (exposure to sunlight for few hours)
of these nitroaromatic can generate small fragment molecules (e.g., malonic acid, glyoxylic
acid) and rapidly reduce the particle absorption (Hems and Abbatt, 2018; Wang et al., 2019b;
Li et al., 2020). The complexity in composition and sources, as well as the dynamics in their
atmospheric processing limit our understanding in BrC chromophores and their links to light
absorption.
In recent years, a growing number of studies have investigated the chromophore
composition of BrC and found that nitro-phenols, low ring acids/alcohols, PAHs and carbonyl





oxygenated PAHs (OPAHs) were the major chromophores in BrC (Teich et al., 2017; Yuan et
al., 2020; Huang et al., 2020). Some chromophores in BrC can be generated from both primary
emission and secondary formation. For example, 4-nitrophenol and 4-nitrocatechol can be
emitted directly from biomass burning and can also be generated through photo-oxidation
reactions (Kitanovski et al., 2012; Yuan et al., 2020). The differences in emission sources or
atmospheric oxidation conditions have a significant effect on the chemical composition of BrC
chromophores. Previous studies mainly focused on seasonal variations of BrC chromophores
(Wang et al., 2018; Kasthuriarachchi et al., 2020; Yuan et al., 2021), and the diurnal variation
of WS-BrC in fluorescence and inorganic fractions (Deng et al., 2022; Zhan et al., 2022),
however, the research of BrC chemical composition on day-night differences is scarce. In this
study, the optical properties and chemical composition of the WS-BrC and WIS-BrC in daytime
and nighttime were measured with a high-performance liquid chromatography–photodiode
array–high-resolution mass spectrometry platform (HPLC-PDA-HRMS) in PM$_{2.5}$ samples
collected in Shijiazhuang, one of the most polluted cities in the Beijing-Tianjin-Hebei region.
Besides, the relationship between the concentration and light-absorbing contributions of the
BrC subgroups was analyzed. The object of this study is to investigate the day-night differences
in the optical properties and chromophore composition of BrC and to explore the effect of
primary emissions and atmospheric processes on the light absorption and chemical composition
of BrC.
**2    Experimental**
**2.1 Sample collection.**

87        Day and night PM$_{2.5}$ samples were collected on the quartz-fiber filters (8*10 in., Whatman,

QM-A; filters prebaked at 750 °C, over 3 h) through a high-volume air sampler (Hi-Vol PM$_{2.5}$
sampler, Tisch, the velocity of flow ~1.03 m$^3$ min$^{-1}$, Cleveland, OH) from 17 January to 13
February 2014. Daytime samples were collected from 08:30 to 18:30 (~10 hours), and nighttime
samples are collected from 18:30 to the next day at 8:30 (~14 hours). After collection, the
samples were stored in a freezer (-20°C) until analysis. The sampling site was located on the
rooftop of a building (~15 m above ground) in the Institute of Genetics and Developmental



Biology, Chinese Academy of Sciences (38.2° N, 114.3° E), which is surrounded by a
residential–business mixed zone.
**2.2 Light Absorption Measurement.**
A portion filter (about 0.526 cm$^2$ punch) was taken from collected samples and sonicated
for 30 min in 10 mL of ultrapure water (>18.2 MΩ) or methanol (J. T. Baker, HPLC grade), and
then the extracts WS-BrC and methanol soluble BrC (MS-BrC) were obtained. The extracts
were filtered with a 0.45 μm PVDF (water soluble) or PTFE (water insoluble) pore syringe
filter to remove insoluble substances. The light absorption spectra of the filtrate were tested
using a UV-VIS spectrophotometer (Ocean Optics) over the range from 250 nm to 700 nm,
equipped with a liquid waveguide capillary cell (LWCC-3100, World Precision Instruments,
Sarasota, FL, USA), following the method of Hecobian et al. (2010). To ensure reliable
absorbance measurements (Absorbance between 0.2 to 0.8 M m$^{-1}$ at 300nm in this study), the
filtrate was diluted with appropriate folds before absorption spectra measurements. In this study,
the light-absorbing of WIS-BrC is obtained by MS-BrC minus WS-BrC. The light absorption
coefficient (Abs) and absorption data were calculated following the equation:
$$Abs_\lambda = (A_\lambda - A_{700}) \frac{V_l}{V_b \times l} \times \ln(10)$$
where Abs$_\lambda$ (M m$^{-1}$) represents the sample absorption coefficient at wavelength of λ; A$_\lambda$ is the
absorbance recorded (Random wavelength); A$_{700}$ for explaining baseline drift as the reference
during data analysis. V$_l$ (ml) is the total volume of solvent (water or methanol) used to extract
the quartz-fiber filters; V$_b$ (m$^3$) represents the volume of through the filter sample of the air; l
(0.94 m) is the optical path length of UV-VIS spectrophotometer and ln (10) is the absorption
coefficient with base-e, which is the natural logarithm by using the logarithm conversion with
the base-10.
About the mass absorption efficiency (MAE) of the filter extracts at wavelength of λ can be
defined as:
$$MAE_\lambda = Abs_\lambda / C_{OM}$$
where C$_{OM}$ (μg m$^{-3}$) stands for the concentration of water-soluble organic carbon (WSOC) or
methanol extracts methanol-soluble organic carbon (MSOC). The concentrations of WSOC
were measured with a TOC-TN analyzer (TOC-L, Shimadzu, Japan). The concentration of OC



was measured by a thermal-optical carbon analyzer (DRI, Model 2001) with the IMPROVE A
protocol (Chow et al., 2011). Note that MSOC is usually replaced with OC because previous
studies have shown that methanol has a high extraction efficiency (~90%) for OC. But it is
difficult to completely extract the OC by methanol (Chen and Bond, 2010; Cheng et al., 2016;
Xie et al., 2019). Here, WISOC is obtained by MSOC minus WSOC.
The wavelength dependence that the light absorption chromophore of solution can be
characterized by this equation:
$$Abs_\lambda = K \cdot \lambda^{-AAE}$$
where K is the fitting parameter of the extracts which is constant related to the chromophoric
concentration; AAE is known as the absorption Ångström exponent, which depends on the
types of chromophores in the solution. In this study, AAE was calculated by linear regression
of $\log_{10} Abs_\lambda$ versus $\log10_\lambda$ at 300–400 nm.
The MAE of standards samples ($MAE_{S,\lambda}$), e.g., 4-nitrocatechol and 4-nitrophenol, in the
water or methanol solvent at a wavelength of λ were calculated as the Laskin et al. (2015)
$$MAE_{S,\lambda} = \frac{A_\lambda - A_{700}}{l \times C} \ln(10)$$
where C (µg mL$^{-1}$) is the concentration of the standards in the extracts.
**2.3 BrC Chemical composition analysis.**
The main chromophores in WS-BrC and WIS-BrC were identified by the HPLC-PDA-
HRMS platform (Thermo Electron, Inc.), and the details are presented in our previous study
(Huang et al., 2020). Firstly, the filter samples (3.5~48.3 cm$^2$) were ultrasonically extracted
with 6 mL of the ultrapure water for 30 min and repeated two times. The extracts were filtered
through a PVDF filter (0.45 µm) to remove insoluble materials. Then the solution was subjected
to an SPE cartridge (Oasis HLB, USA) to remove water-soluble inorganic salt ions. On the
other hand, the residual filters were dried and the WIS-BrC fractions were further extracted two
times with 6 mL of methanol for 30 min. to extract the WIS-BrC fractions. Afterward, the
extracts of WS-BrC and WIS-BrC chromophores were dried with a gentle stream of nitrogen
and then redissolved in 150 µl of ultrapure water and methanol.
The BrC factions were analyzed by an HPLC-PDA-HRMS platform (including the Dionex
UltiMate system and the high-resolution Q Exactive Plus hybrid quadrupole-Orbitrap mass



spectrometer). Here, the extracts were loaded onto a Thermo Accucore RP-MS column by the
binary solvent with an aqueous solution containing 0.1% formic acid and methanol solution
containing 0.1% formic acid as mobile phases $L_1$ and $L_2$, eluting at a flow rate of 0.3 mL min$^{-1}$
$^1$. The process of gradient elution here was set as follows: firstly, aggrandize linearly the
concentration of $L_2$ from 15% to 30% in the preliminary 15 minutes, and then linearly increased
to 90% from 15 to 45minutes, held at 90% from 45 to 50 minutes, afterward decreased to 15%
from 50 to 52 minutes and held there for 60 minutes. The Q Exactive Plus hybrid quadrupole-
Orbitrap mass spectrometer, negative/positive mode ESI (-)/ESI (+) for details usage and data
processing can refer to in the article by Huang et al. (2020) and Liu et al. (2016). Briefly,
HPLC/PDA/HRMS platform was employed in ESI (-) and ESI (+) mode to acquire BrC
fractions that mass range from m/z 100 to 800. Strongly polar aromatic hydrocarbons like
nitrophenol and carboxylic acid are preferentially ionized in ESI (-) mode, conversely, ESI (+)
mode is helpful to detect OPAHs (Oxygenated PAHs) and PAHs fractions (Lin et al., 2017).
The absorption spectra of chromophores were measured by a PDA detector in the wavelength
range of 190-700 nm. In this study, 38 BrC components (20 WS-BrC and 18 WIS-BrC species)
were detected by mass spectrometry and PDA spectroscopy (see Table S1). Mass data processed
by the Xcalibur 4.0 software which the parameter of molecular mass set in ± 3 ppm and
maximum numbers of atoms for the formula calculator set as 30 $^{12}$C, 60 $^1$H, 15 $^{16}$O, 3 $^{14}$N, 1 $^{32}$S,
1 $^{23}$Na. The results of this study were corrected by a blank.

## 3 Results and discussion

### 3.1 Optical properties of BrC during the day and night.

Figure 1 (a) shows the average absorption spectra of WS-BrC and WIS-BrC at the
wavelength range between 300 and 500 nm during the day and the night. It can be seen that the
light absorption of both WS-BrC and WIS-BrC sharply increased toward the short wavelength.
The average absorbance of WS-BrC is 46.04 ± 35.92 M m$^{-1}$ (at 365 nm) during the day that is
much higher than the night (27.90 ± 24.80 M m$^{-1}$). However, the light absorption of WIS-BrC
at 365 nm is lower during the night (35.68 ± 35.50 Mm$^{-1}$) than during the day (40.89 ± 23.42
M m$^{-1}$). The day-night differences of light absorption of WS-BrC and WIS-BrC indicates the





difference in water solubility and polarity of the chromophores. The average AAE of WS-BrC
($AAE_{WS\text{-}BrC}$) and WIS-BrC ($AAE_{WIS\text{-}BrC}$) during the day are $5.10 \pm 0.28$ and $6.36 \pm 0.45$,
respectively, which are lower than those of the night ($5.51 \pm 0.40$ and $6.97 \pm 0.80$, respectively).
Note that both during the day and night the $AAE_{WS\text{-}BrC}$ is lower than $AAE_{WIS\text{-}BrC}$, which is
different from findings in previous studies (see Table S2). For example, Huang et al. (2020)
found that the $AAE_{WS\text{-}BrC}$ was higher ($8.2 \pm 1.0$ and $8.2 \pm 1.0$ in Beijing and Xi'an, respectively)
than that of $AAE_{WIS\text{-}BrC}$ ($5.7 \pm 0.2$ and $5.4 \pm 0.2$ in Beijing and Xi'an, respectively). Besides,
$MAE_{365}$ of WS-BrC are 2.0- and 2.5-fold of WIS-BrC during the day ($2.88 \pm 0.24$ vs. $1.43 \pm$
$0.83$ $m^2\,g\,C^{-1}$) and night ($2.58 \pm 0.14$ vs. $1.02 \pm 0.49$ $m^2\,g\,C^{-1}$), respectively, which is opposed
to the results of previous studies. For example, the $MAE_{365}$ of WS-BrC are 0.7- and 0.5-fold of
WIS-BrC in winter of Beijing ($1.22 \pm 0.11$ vs. $1.66 \pm 0.48$ $m^2\,g\,C^{-1}$) (Chen and Bond, 2010)
and Xi'an ($1.00 \pm 0.18$ vs. $1.82 \pm 1.06$ $m^2\,g\,C^{-1}$) (Li et al., 2020), respectively. This result
indicates that the chemical composition of BrC in the most polluted city, Shijiazhuang, is
different from other urban areas on primary sources and secondary aging process. However,
both WS-BrC and WIS-BrC have higher $MAE_{365}$ and average AAE values during the day than
the night. This suggests that the day-night differences of AAE and $MAE_{365}$ of BrC fractions are
likely associated with the different primary emissions and atmospheric aging processes (Cheng
et al., 2016; Wang et al., 2019a; Wang et al., 2020). For example, the AAE and $MAE_{365}$ of BrC
emitted from biomass burning (AAE ~7.31, and $MAE_{365}$ ~1.01 $m^2\,g\,C^{-1}$, respectively) (Siemens
et al., 2022) showed large differences with that from vehicle emissions (AAE ~10.5, and
$MAE_{365}$ ~0.32 $m^2\,g\,C^{-1}$) (Xue et al., 2018). Besides, photochemical oxidation of fresh BrC from
coal combustion resulted in considerable changes in AAE and $MAE_{365}$, e.g., the AAE and
$MAE_{365}$ of fresh coal combustion emission are 7.2 and 0. $84 \pm 0.54$ $m^2\,g\,C^{-1}$, much higher than
those in aged samples (6.4 and $0.14 \pm 0.08$ $m^2\,g\,C^{-1}$, respectively) (Ni et al., 2021).

204         Figure 1 (b) shows the light absorption contributions of WS-BrC and WIS-BrC to total

BrC over the wavelength range of 300–500 nm. It is obvious that the absorption contribution
of WS-BrC is increased from 53.8% at 300 nm to 87.4% at 500 nm during the day, and from
38.4% to 61.5% during the night. The higher absorption contributions of WS-BrC at longer
wavelengths during the day compared to that of the night may be related to photo-oxidation



reaction in day time (Wang et al., 2019b; Chen et al., 2021). The absorption contribution of
WS-BrC accounts for 62 ± 8% to total BrC absorption at 365 nm during the day, but only 47 ±
8% during the night. The large difference in BrC light absorption between samples from the
day and those from the night observed in this study is comparable with previous studies (Shen
et al., 2019; Li et al., 2020), and indicates the significant day-night difference in chemical
composition.
**3.2 Composition and absorption contribution of BrC during day and night**.
In total, 38 major chromophores were quantified in WS-BrC and WIS-BrC with HPLC-
PDA-HRMS analysis. According to the characteristics of the molecular structures and
absorption spectra, these chromophores are divided into ten subgroups, including two
quinolines, four nitrocatechols, six nitrophenols, four aromatic alcohols/acids, four 2–3-ring
OPAHs, three 4-ring OPAHs, two 3-ring PAHs, four 4-ring PAHs, five 5-ring PAHs, and four
6-ring PAHs. Detailed information about these chromophores is listed in Table S3. Figure 2
shows the chemical composition of the identified BrC components during the day and night.
The total concentration of these chromophores during the day (169.8 ng/m$^3$) is similar to that
at night (171.8 ng/m$^3$), and the chemical composition of the BrC subgroups is clearly different
between the day and night. For example, nitrocatechols, aromatic alcohols/acids and 2–3-rings
OPAHs are the major contributors to the total mass concentration of identified BrC
chromophores during the day (accounting for 23.3%, 22.3%, and 16.6%, respectively). These
BrC chromophores, however, are the minor components during the night (accounting for 12.1%
and 15.6%, and 6.9%, respectively). This result indicates the enhanced formation of these
chromophores during the day. On the contrary, the relative contributions of nitrophenols and 4–
6-ring PAHs are much lower during the day (15.3% and 15.2%, respectively) than those during
the night (35.8% and 24.0%, respectively). During the night, 4-nitrophenol (4NP) contributes
24.4% of the total concentration, followed by 2-methyl-4-nitrophenol, fluoranthene, and
chrysene (2M4NP 4.7%, FLU 4.6%, CHR 4.6%, respectively). The higher contributions of
nitrophenols and 4–6-rings PAHs at night are likely caused by enhanced primary emissions (Lin
et al., 2020; Chen et al., 2021). Our previous study has found that the emitted organic aerosols
from coal combustion had a clearly increase at midnight in Shijiazhuang (Huang et al., 2019;


Lin et al., 2020). Thus, the large contribute of nitrophenols and 4–6-rings PAHs to total mass
concentration at night that may be impacted by emissions from the coal combustion.

240         To investigate the source of the BrC chromophores, the mass concentrations (these

concentrations of chromophores are OC normalized) of the day and night were compared. The
day-to-night ratios of identified BrC compounds in mass concentrations is shown in Figure 3.
It can be seen that the average day-to-night ratios of WS-BrC chromophores are 4.87 for
quinolines, 3.49 for 2–3-ring OPAHs, 3.47 for nitrocatechols, 0.48 for nitrophenols, and 2.53
for aromatic alcohols/acids, respectively. Previous studies have found that quinolines are
important products of fossil fuel combustion, and were used as tracers of the vehicular exhaust
(Banerjee and Zare, 2015; Xue et al., 2018; Lyu et al., 2019). Thus, the higher day-to-night ratio
of quinolines may be due to increased primary emissions from vehicles during the day. Nitro-
phenols and vanillin are typical biomass burning tracers for atmospheric aerosols (Harrison et
al., 2005; Scaramboni et al., 2015; Huang et al., 2021). Previous studies have identified
econdary formation as an important source of phthalic acid (PA) and methyl-nitrocatechols
(Chow et al., 2015; Zhang and Hatakeyama, 2016; Liu et al., 2017). In this study, vanillin,
phthalic acid, and three methyl-nitrocatechols (including 4M5NC, 3M6NC, and 3M5NC)
isomers have high day-to-night ratios (4.16, 3.75 and 3.28, respectively). The high day-to-night
ratios of these BrC chromophores suggest that biomass burning and secondary formation likely
play important roles in the daytime source of BrC.

257         However, the average day-to-night ratio (~0.48) of nitrophenols is smaller than one. The

day-to-night ratio of nitrophenols is similar to values in previous studies (Yuan et al., 2016;
Schnitzler and Abbatt, 2018). Besides, previous studies found that emissions from residential
coal-fired heating are significant sources of nitrophenols (Wang et al., 2018; Lu et al., 2019).
This suggests that coal combustion emissions have important effect on the nocturnal
concentration of nitrophenols. Compared with the WS-BrC chromophores (the day-to-night
ratio > 2.53), the day-to-night ratios of the WIS-BrC chromophores approach or below one,
with average ratios of 1.46 for 3-ring PAHs, 1.34 for 4-ring OPAHs, 0.74 for 4-ring PAHs, 0.91
for 5-ring PAHs, and 0.79 for 6-ring PAHs, respectively. A number of studies showed that coal
combustion was the dominant source of PAHs (Wang et al., 2018; Xie et al., 2019; Yuan et al.,



2020). Thus, the local emissions may be responsible for the majority of 4–6-rings PAHs during
the night.
Figure S1 (see Supplemental Information) shows the light absorption contributions of the
BrC subgroups to total BrC subgroups in the wavelength range between 300 and 420 nm (the
absorptions above 420 nm are too low to exactly estimate the contributions), exhibiting large
day-night difference. For example, quinolines show evident absorption below 340 nm (3.6% at
310 nm during the day), but negligible contribution above 360 nm. Nitrophenols exhibit a
maximum contribution at about 350 nm, while nitrocatechols show higher absorption in the
wavelength range from 360 to 400 nm. For PAHs, the absorption maxima shift to a longer
wavelength with the increase of the aromatic rings (e.g., 320 nm for 4-ring PAHs and 400 nm
for 6-ring PAHs). Overall, the combined light absorption contributions of nitrophenols,
nitrocatechols, and PAHs are 86.5% and 80.1% (averaged between 300 and 420 nm) at night
and day, respectively. This result is similar to previous studies, in which PAHs and nitro
aromatic compounds were identified as the major chromophores (Huang et al., 2020; Yuan et
al., 2020).
The light absorption contribution of these BrC subgroups exhibits obvious day-night
differences. For example, the absorption contribution of 2–3-rings OPAHs and nitrocatechols
at 365 nm increased by ~2.0 and ~3.5 times during the day compared to that during the night
(see Figure S2). This result differs from previous studies (Kampf et al., 2012; Gao et al., 2022),
which indicated that light absorption of BrC compounds were enhanced after exposure to photo-
oxidation. On the other hand, the absorption contributions of nitrophenols and 4–6-rings PAHs
at 365 nm are ~1.6 times and ~2.2 times higher at night than at day, respectively. The day-night
difference of light absorption of nitrophenols is comparable with previous studies (Harrison et
al., 2005; Wang et al., 2020). High absorbance of nitrophenols at night is closely related to their
higher mass fraction at night. The absorption characteristics of 4–6-ring PAHs are significantly
different from the nitro-phenols, and their absorption per unit mass is larger than that of nitro-
phenols. The per unit mass absorbance of PAHs much higher than the low-ring aromatic
hydrocarbons (e.g., aromatic alcohols/acids) are due to their strongly conjugated systems. It is
worth noting that the absorption contributions of some BrC compounds (including quinolines,





aromatic alcohols/acids, 4-ring OPAHs, 3-ring PAHs four subgroups) are much lower than
those of the above-mentioned BrC compounds because of their lower mass concentration or
light absorption coefficient.

**3.3 Comparisons between the low and high pollution period.**

The relative contributions of day-night subgroups of BrC chromophores in light absorption
and mass concentration were further investigated for different pollution levels. The sampling
campaign was classified into low-pollution period ($PM_{2.5}$< 150 µg m$^{-3}$) and high-pollution
period ($PM_{2.5}$ >250 µg m$^{-3}$). Figure 4 (a) shows the mass fractional contributions of the
identified subgroups during these periods, which show an evidently different during the day
and night. For example, the mass fraction of quinolines during the day (~24.1%) is much higher
than during the night (3.4%) at low-pollution period, which may be related to increased vehicle
emissions at day (Rogge et al., 1993; Lyu et al., 2019). Moreover, during the low-pollution
period, with good atmospheric dispersion conditions during the day, the fractional concentration
of BrC is only 56.9 ng m$^{-3}$ much lower than the nights and high-pollution periods. In the high-
pollution period, however, the mass concentration of quinolines is much lower than other BrC
chromophores and there is no evident difference between day and night. The mass fraction of
aromatic alcohols/acids during the day (35.4%) is much higher than during the night (12.0%)
at low-pollution period. For high-pollution period, the mass fraction of aromatic alcohols/acids
shows little difference between the day and night. However, their mass concentration during
the day (55.5 ng m$^{-3}$) is higher than that during the night (31.9 ng m$^{-3}$). Thereinto, the mass
concentration of phthalic acid (a tracer from photochemical oxidation) contributes more than
60% to the aromatic alcohols/acids at day for low and high-pollution period (Zhang and
Hatakeyama, 2016). This evidence may be suggesting that there is stronger photo-chemical
oxidation for aromatic alcohols/acids during the day, especially at low-pollution period.
The mass fractional contribution of nitrocatechols is lower during the day than the night
at low-pollution period, while there is obvious secondary formation during the day for high-
pollution period. This likely suggests that the daytime conditions of the high-pollution period
are inducive for the generation of nitrocatechols. The mass fractional contribution of PAHs
during the day is much lower than the night at low-pollution period. At night, residential coal





heating is an important source of PAHs, and therefore the daytime contributions of PAHs are
much lower than nighttime (Wang et al., 2017; Ni et al., 2021). While there is no day-night
difference for PAHs at high-pollution period, which is related to the stable sources and stagnant
weather conditions (Huang et al., 2019; Lin et al., 2020). It is noteworthy that the mass
contributions of the nitrophenols (nighttime is 2–3 times more than daytime) and 2–3-rings
OPAHs (daytime is ~2 times more than nighttime) is opposite between the day and night. This
demonstrates that they have stable sources compared to other BrC subgroups even during the
low-pollution period and high-pollution period. The higher mass fractional contribution of 2–
3-rings OPAHs during the day is related to photochemical oxidation. Nitrophenols exhibit a
higher mass fractional contribution during the night than the day, indicating a significant
contribution from primary emissions (Lu et al., 2019; Lin et al., 2020). Besides, previous
investigations have shown that $NO_x$ concentrations and relative humidity are higher at night in
Shijiazhuang, which may have accelerated the formation of nitrophenols in the dark (Yuan et
al., 2016; Huang et al., 2019). This result exhibits a clear day-night difference during the low-
pollution period than high-pollution period, which indicates that the low-pollution period is
easily influenced by the external environment (e.g., solar radiation and wind speed).

341       The day-night light absorption contribution of WS-BrC and WIS-BrC chromophores in

different pollution periods is shown in Figure 4 (b). For the low-pollution period, the light
absorption contribution of the ten BrC subgroups shows a large difference during the day and
night. Thereinto, the WS-BrC chromophores (e.g., quinolines, nitrophenols and nitrocatechols)
is the main contributor (accounting for ~75% at 365 nm) of total identified BrC during the day.
While, the WIS-BrC chromophores (e.g., 4–6-rings PAHs) become an abundance contributor
(accounting for ~65% at 365 nm) during the night. There is an obvious day-night differences in
light absorption at low-pollution period, which is consistent with the difference in their mass
concentration contribution. Different from the low-pollution period, the light absorption
contribution of the total WS-BrC and WIS-BrC chromophores showed no significant day-night
differences during the high-pollution period. However, the absorption contributions of
subgroups in WS-BrC chromophores have a significant day-night difference (e.g., nitrocatechol
and nitrophenols) during the high-pollution period, which is due to the change of the mass





contributions. WS-BrC chromophores have stronger light absorption both during the day and
night compared to the WIS-BrC chromophores at high pollution period. Specifically, the
absorption contribution of nitrocatechols and nitrophenols combined accounts for 66.1% at day
and 60.7% at night at 365 nm, respectively, which depend on the different emission sources or
formation mechanisms between during the day and night. Our results show a significant day-
night differences in mass contributions and absorption contributions of BrC components at
different pollution levels. This suggests that the variation of BrC chromophores in different
pollution periods may be caused by different sources and weather conditions.
**4   Conclusions**
In general, our study shows the large day-night differences in optical properties and
chemical composition of the bulk BrC in urban atmosphere. Thereinto, WS-BrC is the main
light-absorbing contributors during the day, while WIS-BrC is main light-absorbing compound
at night. Different types of the identified BrC chromophores exhibit unique characteristics of
day-night differences, reflecting their particular sources and formation pathways. For example,
nitrocatechols and 2–3-rings OPAHs are important contributors to mass concentration and light
absorption during the day, while 4–6-rings PAHs and nitrophenols become the significant
contributors at night. Day-night differences of BrC chromophores are associated with different
sources during day (more secondary formation) and night (more primary emissions).
Moreover, these day-night differences are largely affected by the air pollution level, which
determines the concentrations of BrC precursors (e.g., aromatic hydrocarbon and phenols) and
oxidants (e.g., $NO_x$, $NO_3^{\cdot}$ and OH), as well as meteorological conditions (e.g., solar irradiation
and RH) (Liu et al., 2012; Laskin et al., 2015; Wang et al., 2019). For example, our results found
that the day-night difference of BrC fractions is more pronounced in chemical composition and
light absorption during the low-pollution period than high-pollution period. These factors may
show different effects on the formation and photobleaching of different types of the identified
chromophores. However, our current understanding of the formation mechanisms of and
influencing factors on these identified chromophores is still incomplete (Huang et al., 2018;
Yuan et al., 2020). Therefore, a combination of more laboratory and field studies is needed to
(1) make comprehensive characterization of the chromophore composition BrC in ambient



aerosol; (2) explore thoroughly the formation mechanisms of different types of BrC
chromophore. This will significantly enhance our understanding of atmospheric BrC formation
mechanisms and therefore improve the accuracy of the atmospheric effects of BrC in air quality
and climate models.





**Data availability.** Detailed data can be obtained from https://doi.org/10.5281/zenodo.7690230.
**Author contributions**. Ru-jin Huang designed the study. Data analysis was done by Yuquan Gong
and Rujin Huang. Yuquan Gong and Rujin Huang interpreted data, prepared the display items and
wrote the manuscript. Lu Yang, Ting Wang, Wei Yuan, Wei Xu, Wenjuan Cao, Yang Wang, and
Yongjie Li commented on and discussed the manuscript.
**Competing interests.** The authors declare that they have no conflict of interest.
**Acknowledgements.** We are very grateful to the various grants that supported this study, and
we also appreciate each co-author's comments and help.
**Financial support.** This work was supported by the National Natural Science Foundation of
China (NSFC) under Grant No. 41925015, the Strategic Priority Research Program of Chinese
Academy of Sciences (No. XDB40000000), the Chinese Academy of Sciences (No. ZDBS-LY-
DQC001), and the Cross Innovative Team fund from the State Key Laboratory of Loess and
Quaternary Geology (No. SKLLQGTD1801).



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





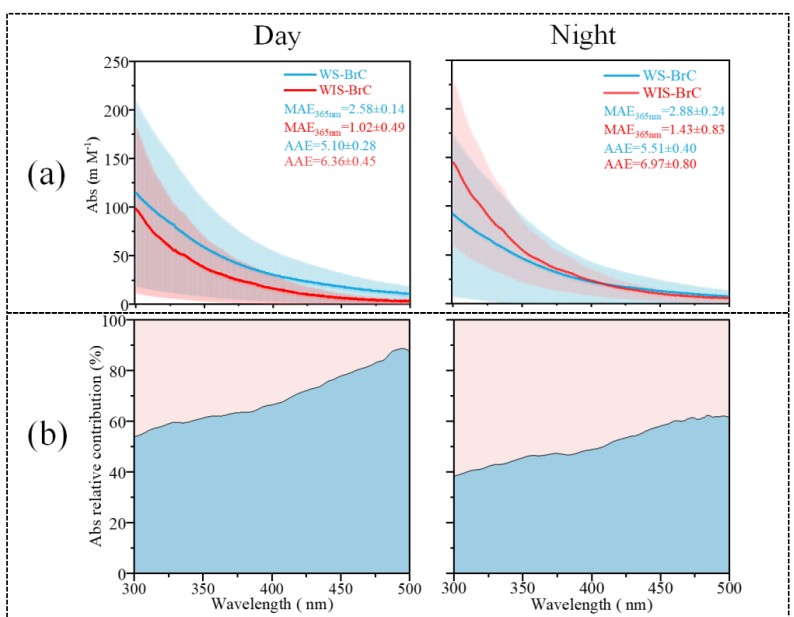

**Figure 1.** (a) Day-night absorption spectra (Abs, in the wavelength range of 300–500 nm), mass absorption efficiency (MAE, determined at 365 nm), and absorption Ångström exponent (AAE, calculated between 300 and 400 nm) of water-soluble/insoluble BrC (WS-/WIS-BrC) in Shijiazhuang. (b) Light-absorbing proportion of WS-BrC and WIS-BrC between 300 to 500 nm.

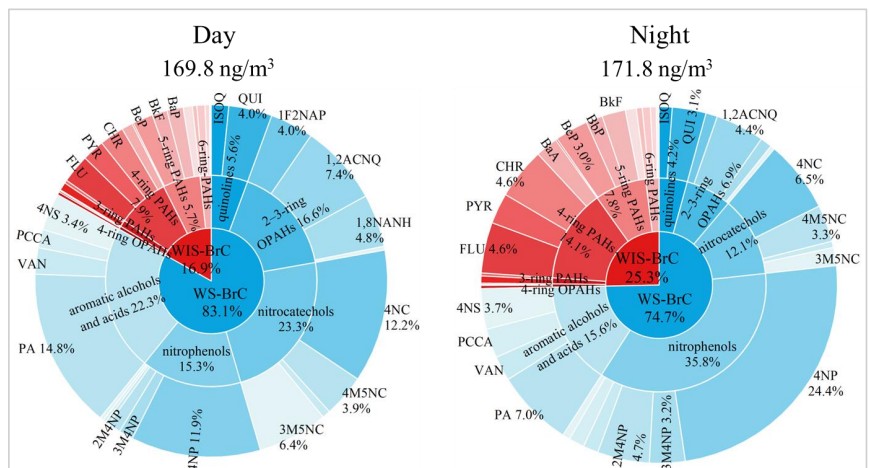

**Figure 2.** Mass fraction of the identified BrC chromophores during the day and night (details of the identified BrC chromophores are shown in Table S1).

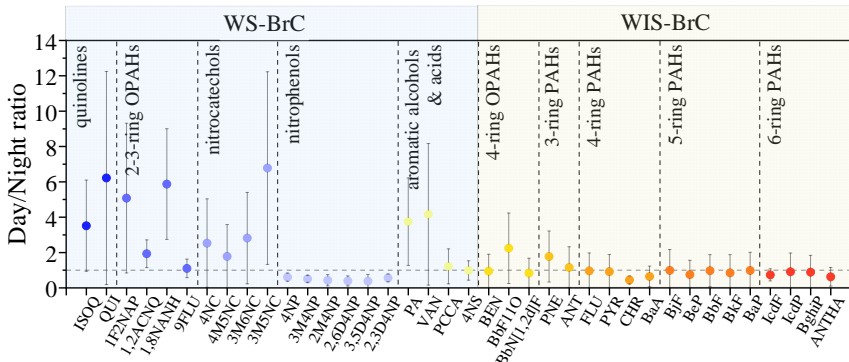

**Figure 3**. Day-to-night ratios of the concentrations of different BrC chromophores.

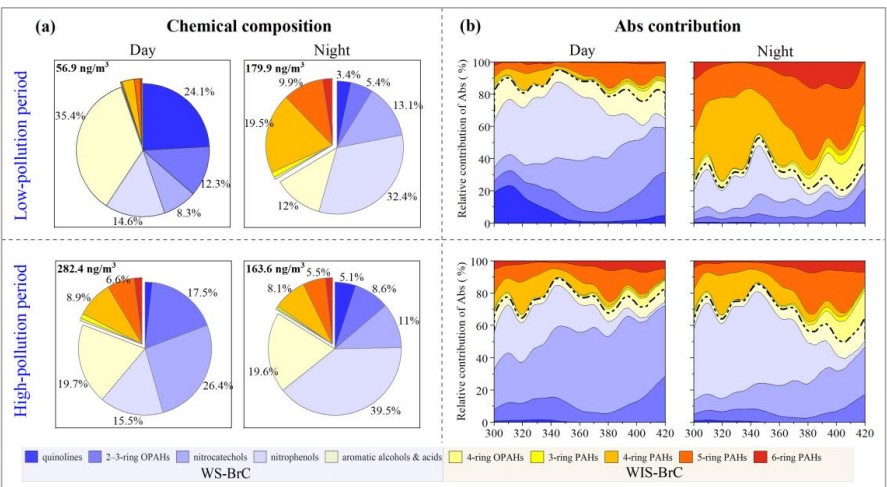

**Figure 4.** Day-night fractional contributions of mass concentrations **(a)** and light absorption **(b)** of the ten BrC subgroups in low-pollution period and high-pollution period. Here the BrC chromophore is the main chromophore substance that has been identified. In **(b)** WS-BrC is below the dotted line and WIS-BrC is above the dotted line.