# Peer review of "Yuquan Gong1,2, Ru-Jin Huang1,2,3, Lu Yang1,2, Ting Wang1, Wei Yuan1,2, Wei Xu1, Wenjuan Cao1, Yang Wang4,5, Yongjie Li6"

_Atmospheric Chemistry and Physics, 2023_

## Author Response (AR1)

The authors thank the editor and referees to review our manuscript and particularly for the valuable comments and suggestions that are very helpful in improving the manuscript. We provide below point-by-point responses to the referees' comments (in blue). We also have made most of the changes suggested by the referees in the revised manuscript.

**Referee #1**

In this manuscript, the authors present a study of the light absorption and composition of water-soluble and -insoluble brown carbon (BrC) from filter samples collected in a highly polluted urban environment. The sample preparation and the spectroscopic and mass spectrometric analyses are carefully performed. The observations during the day and night are thoroughly discussed and support insights into the emission and evolution of BrC. For example, more polar, water-soluble components are more abundant during the day, when photochemical aging likely drives the functionalization of primary emissions. I think the manuscript is suitable for publication as a measurement report in ACP after minor technical revisions.

Response: We thank the referee for positive comments.

105 - Absorbance is unitless.

Response: The unit has been deleted.

107 - What are the potential interferences for this subtraction (e.g., solvent and pH effects)? Was the pH of the water extracts measured or adjusted?

Response: The potential interferences of this subtraction is estimated by comparing the absorbance of the same filter sample extracted by sequential extraction with water (WS-BrC) and methanol (WIS-BrC) and by direct extraction with methanol (MS-BrC). We have added this point in the revised manuscript lines 108-113 and it reads "As shown

in Figure S1, the summed absorbance of WS-BrC and WIS-BrC is very close to the absorbance of MS-BrC (difference less than 5%). Therefore, the interferences of solvent and pH on the measurement of WIS-BrC should be very limited. The pH of the water extracts was not adjusted because highly diluted water extracts was used to measure the light absorption, and little change of pH was observed for water extracts of different samples."

[Figure]

**Figure S1.** Comparison of the UV-Vis spectra of BrC extracts between sequential extraction with water and methanol and direct extraction with methanol.

110 - I think it is clearer to write the units of Abs without a space (i.e., Mm^-1) here and throughout. In Figure 1a, the units of Abs also need to be corrected from m M^-1.

Response: Change made.

111 - Is there a precedent for the subtraction of A_700? If not, a brief justification should be provided here (e.g., BrC does not absorb at such long wavelengths, so a shift can be attributed to a change in the photons reaching the waveguide capillary cell).

Response: Yes, previous studies have used this protocol to correct for the baseline drift when measuring the light absorption with LWCC (Hecobian et al., 2010; Huang et al., 2020). We have added one sentence in the revised manuscript in line 118-120: "To account for baseline drift that may occur during analysis, absorption at all wavelengths below 700 nm are referenced to that at 700 nm where there is no absorption for BrC

extracts."

166 - Standards, including 4-nitrocatechol and 4-nitrophenol, were used in the spectroscopic analysis - were standards also used in the MS analysis? In other words, how were the concentrations of the 38 identified components determined?

Response: In this study, 28 chromophores are quantified by authentic standards and 10 chromophores are quantified by surrogates with similar structures (see Table S1 in Supplemental Information). Thereinto, the WS-BrC chromophores (1#−20#) and water-insoluble 4-ring OPAHs (including 21#, 22#) are quantified by MS analysis, and the rest WIS-BrC chromophores (23#−26#, 27# and 28#−38#, i.e., PAHs) are quantified by spectroscopic analysis due to their super low ionization efficiency in ESI. We have added this point in the revised manuscript in lines 185-193 and it reads "In total, 20 WS-BrC chromophores (two quinolines, four 2-3 ring OPAHs, four nitrocatechols, six nitrophenols and four aromatic alcohols & acids) and 18 WIS-BrC chromophores (three 4-ring OPAHs and 15 PAHs) were identified and their concentrations were quantified with authentic standards (28 species) or surrogates (10 species) (see Table S1). Thereinto, the WS-BrC chromophores, benzanthrone (21#) and benzo[b]fluoren-11-one (22#) were quantified by mass spectrometry analysis in either negative or positive ESI mode, while the rest of WIS-BrC chromophores were quantified by PDA spectroscopic analysis due to their super low ionization efficiency in ESI (see Table S1)."

230 - I think a clearer explanation of the difference between nitrophenols and nitrocatechols should be presented. Both can be emitted from biomass burning, for example, so it may be surprising that they differ so much throughout this study.

Response: Thanks for pointing it out. We have added explanation in the revised manuscript. In lines 297-304, it now reads "Although both nitrophenols and nitrocatechols can be emitted from biomass burning, they show largely different daynight variation patterns. The higher concentrations of nitrocatechols during daytime indicate enhanced secondary formation, which is similar to the results observed in urban Beijing (Cheng et al., 2021). In addition, previous studies found that emissions from residential coal-fired heating are significant sources of nitrophenols (Wang et al., 2018; Lu et al., 2019). The higher concentrations of nitrophenols during nighttime, however, suggest that they are mainly emitted from primary emission sources such as residential heating during winter in North China."

365 - A more direct connection from polarity and absorptivity to functionalization could be incorporated here or earlier.

Response: We have added this point in revised manuscript in lines 414-415, it now reads "The polar WS-BrC has higher $MAE_{365}$ compared to the less-polar WIS-BrC, mainly due to the different conjugate systems and functional groups in the two fractions."

371 - Some primary emissions are also reduced during the night (e.g., transportation emissions). Are these changes reflected in the total mass concentrations or elsewhere?

Response: We agree with the referee that some primary emissions, such as vehicle emissions, are reduced during the night, which results in the differences in BrC chromophore composition between day and night. For example, the concentrations of isoquinoline and quinoline, two chromophores mainly emitted from vehicle exhaust (Banerjee and Zare, 2015; Lyu et al., 2019), decreased from 9.5 ng m$^{-3}$ during daytime to 7.3 ng m$^{-3}$ during nighttime. These results clearly show the effects of primary emissions on the day-night differences in BrC chromophore composition. We have corrected this expression in lines 423-425 and it now reads "Day-night differences of BrC chromophores are associated with different sources during day (mainly secondary formation and vehicle emission) and night (mainly emissions from residential heating) as well as the dynamic development of planetary boundary layer height."

394 - It is unusual to thank the co-authors. Their contributions are acknowledged by their authorship.

Response: Change made.

**Referee #2**

This paper attempts to identify the chromophores in brown carbon and show their diurnal patterns. While there are many very nice aspects of the paper, I am deeply concerned about the big undefined assumptions that are made to link a high-resolution mass to specific structures. Specifically, figure 3 includes the names of specific compounds that are traditionally analyzed using analytical standards however no description is given to describe that sort of targeted analysis. First, when 5 elements are allowed with a 3 ppm mass error for masses ranging up to 800 u, we can expect to have multiple plausible molecular formulas per mass measurement that meet those criteria. So as a first step, we need to know how the plausible formulas were evaluated. Then, perhaps we can calculate elemental ratios and determine the number of unsaturations, however, even the number of unsaturations will require an assumption about the oxidation state of N and S which should be defined. Second, structural analysis and structure confirmation require very deep study using MS/MS and NMR without specific analytical standards. Then we would still want the structures to be defined as tentative until they can be confirmed with specific analytical standards.

Response: Thank you very much for the valuable comments. In this study, 38 chromophores are identified and quantified with authentic standards (28 species) or surrogates (10 species) (see Table S1). Thereinto, the WS-BrC chromophores (1#−20#) and water-insoluble 4-ring OPAHs (including 21#, 22#) are quantified by MS analysis, and the rest WIS-BrC chromophores (23#−26#, 27# and 28#−38#, i.e., PAHs) are quantified by spectroscopic analysis due to their super low ionization efficiency in ESI. We have added this point in the revised manuscript lines 185-193 and it reads "In total, 20 WS-BrC chromophores (two quinolines, four 2-3 ring OPAHs, four nitrocatechols, six nitrophenols and four aromatic alcohols & acids) and 18 WIS-BrC chromophores (three 4-ring OPAHs and 15 PAHs) were identified and their concentrations are quantified with authentic standards (28 species) or surrogates (10 species) (see Table S1). Thereinto, the WS-BrC chromophores, benzanthrone (21#) and benzo[b]fluoren-11-one (22#) were quantified by mass spectrometry analysis in either negative or positive ESI mode, while the rest of WIS-BrC chromophores were quantified by PDA

spectroscopic analysis due to their super low ionization efficiency in ESI (see Table S1)."

We agree with the referee that multiple plausible molecular formulas can meet the criteria of 3 ppm mass error when the masses are high enough. For example, for the m/z of 199.0389 in ESI+ mode (15# chromophore in Table S1), four candidates meet the criteria of 3 ppm: $C_{12}H_7O_3$(-delta=0.355 ppm), $C_5H_{10}O_2N_3SNa$ (delta=1.540 ppm), $C_{13}H_6NNa$ (delta=-1.736 ppm) and $C_4H_{11}O_5N_2S$ (delta=2.921 ppm). To eliminate the chemically unreasonable formulas, these formulas were constrained by the following settings: $0.3 \leq H/C \leq 3.0$, $0.0 \leq O/C \leq 3.0$, $0.0 \leq N/C \leq 1.3$, $0.0 \leq S/C \leq 0.8$ in ESI+ mode as suggested in a previous study (Lin et al., 2012). Further, calculated neutral molecular formulas that did not fit the nitrogen rule were excluded. With these constraints, only two formulas are reserved, i.e., $C_{12}H_7O_3$ and $C_4H_{11}O_5N_2S$. Further comparisons with authentic standards (retention time, absorption spectrum, and mass spectrum) indicate that this compound should be 1,8-naphthalic anhydride ($C_{12}H_6O_3$). We also agree with the referee that the estimation of the number of unsaturations (DBE) requires the assumption about the oxidation state of N and S. We did not estimate DBE in this study. We have added sentences in the revised manuscript in lines 178-185 to explain this point: "The elemental composition of individual chromatographic peaks was assigned with the molecular formula calculator in Xcalibur 4.0 software using a mass tolerance of ±3 ppm and the maximum numbers of atoms for the formula calculator were set as 30 $^{12}C$, 60 $^{1}H$, 15 $^{16}O$, 3 $^{14}N$, 1 $^{32}S$, and 1 $^{23}Na$. To eliminate the chemically unreasonable formulas, the identified formulas were constrained by setting $0.3 \leq H/C \leq 3.0$, $0.0 \leq O/C \leq 3.0$, $0.0 \leq N/C \leq 0.5$, $0.0 \leq S/C \leq 0.2$ in ESI- mode and $0.3 \leq H/C \leq 3.0$, $0.0 \leq O/C \leq 3.0$, $0.0 \leq N/C \leq 1.3$, $0.0 \leq S/C \leq 0.8$ in ESI+ mode, as suggested in a previous study (Lin et al., 2012). Further, the calculated neutral molecular formulas that did not fit the nitrogen rule were excluded."

Third, I strongly doubt that PAHs can be observed using (+) ESI. PAHs are simply too strongly nonpolar for ESI. In fact, I wouldn't expect them to be in MeOH fractions. I would believe that substituted PAHs could be present, but again that would need to be

confirmed using analytical standards.

Response: We agree with the referee. Indeed, the ionization efficiency of PAHs at ESI (+) is very low. In our study, PAHs were quantified by PDA spectroscopic analysis. We have revised the sentences in lines 173-175, it now reads "OPAHs and nitrogen heterocyclic PAHs were quantified in ESI (+) mode, while PAHs were detected by PDA spectroscopic analysis due to their super low ionization efficiency in ESI."
Regarding PAHs measured in methanol extract, a number of previous studies have reported their existence in methanol extract (e.g., Huang et al., 2020; Sun et al., 2021), as also measured in methanol extract in our study.

In figure 2, the mass fractions of identified compounds are given but I do not recall seeing an explanation of how the analytical features were converted to mass fractions. Again many assumptions were likely made that need to be explicitly described.

Response: The mass concentrations of the BrC chromophores are quantified with authentic standards or surrogates as described above, and the results are shown in Table S3. The mass fraction of different chromophores is their relative mass proportion in the total identified BrC chromophores. In line 250, it now reads "…, and the concentrations of these chromophores are shown in Table S3"

**Table S3.** The concentrations of day and night mass of the 38 identified BrC chromophores.

| Name | Mass concentration (ng m$^{-3}$) | |
|---|---|---|
| | Day | Night |
| Isoquinoline | 2.7 ± 1.0 | 2.0 ± 1.3 |
| Quinoline | 6.7 ± 7.7 | 5.3 ± 4.5 |
| 1-Formyl-2-naphthol | 6.8 ± 5.6 | 1.8 ± 1.2 |
| 1,2-acenaphthylenedione | 12.6 ± 10.8 | 7.6 ± 5.9 |
| 1,8-naphthalic anhydride | 8.2 ± 7.6 | 1.8 ± 1.2 |
| 9-fluorenone | 0.6 ± 0.4 | 0.7 ± 0.4 |
| 4-nitrocatechol | 20.7 ± 18.9 | 11.2 ± 7.7 |
| 4-methyl-5-nitrocatechol | 6.6 ± 6.1 | 5.8 ± 3.3 |
| 3-methyl-6-nitrocatechol | 1.3 ± 0.6 | 1.0 ± 0.4 |

| | | |
|---|---|---|
| 3-methyl-5-nitrocatechol | 10.9 ± 11.0 | 2.8 ± 3.2 |
| 4-nitrophenol | 20.2 ± 14.7 | 41.9 ± 29.4 |
| 3-methyl-4-nitrophenol | 2.2 ± 1.5 | 5.6 ± 3.3 |
| 2-methyl-4-nitrophenol | 2.0 ± 1.5 | 8.1 ± 4.6 |
| 2,6-Dimethyl-4-nitrophenol | 0.6 ± 0.4 | 2.4 ± 1.4 |
| 3,5-Dimethyl-4-nitrophenol | 0.4 ± 0.2 | 2.5 ± 1.3 |
| 2,3-Dimethyl-4-nitrophenol | 0.6 ± 0.5 | 1.2 ± 0.6 |
| Phthalic acid | 25.1 ± 12.9 | 12.1 ± 6.3 |
| vanillin | 4.1 ± 1.6 | 3.8 ± 2.8 |
| p-cis-coumaric acid | 2.9 ± 1.5 | 4.6 ± 2.6 |
| 4-nitrosyringol | 5.8 ± 4.8 | 6.3 ± 3.2 |
| Benzanthrone | 0.5 ± 0.4 | 0.5 ± 0.3 |
| Benzo[b]fluoren-11-one | 0.2 ± 0.2 | 0.2 ± 0.1 |
| Benzo[b]naphtho[1,2-d]furan | 0.1 ± 0.1 | 0.2 ± 0.1 |
| Phenanthrene | 1.3 ± 1.2 | 1.1 ± 0.8 |
| Anthracene | 0.5 ± 0.4 | 0.5 ± 0.4 |
| Fluoranthene | 4.4 ± 4.0 | 7.9 ± 7.8 |
| Pyrene | 3.4 ± 3.1 | 4.8 ± 3.4 |
| Chrysene | 3.8 ± 4.2 | 8.0 ± 5.5 |
| Benzo(a)anthracene | 1.7 ± 1.4 | 3.5 ± 2.0 |
| Benzo(j)fluoranthene | 0.2 ± 0.2 | 0.3 ± 0.1 |
| Benzo(e)pyrene | 3.2 ± 3.0 | 5.2 ± 3.6 |
| Benzo(b)fluoranthene | 2.3 ± 2.1 | 2.6 ± 1.6 |
| Benzo(k)fluoranthene | 2.5 ± 2.5 | 3.7 ± 2.5 |
| Benzo(a)pyrene | 1.4 ± 1.3 | 1.7 ± 1.3 |
| Indeno[1,2,3-cd]fluoranthene | 0.7 ± 0.6 | 0.9 ± 0.6 |
| Indeno(1,2,3-cd)pyrene | 1.3 ± 1.2 | 1.3 ± 0.5 |
| Benzo(g,h,i)perylene | 0.7 ± 0.6 | 0.9 ± 0.7 |
| Anthanthrene | 0.3 ± 0.3 | 0.4 ± 0.3 |

Overall, the paper is quite interesting and there is great interest in the identity of BrC. But, gaps in the scientific process must be filled with concrete steps or explicit rationale must be given for educated assumptions that are probably quite reasonable given the deep knowledge on this topic.

Response: Thank you again for the valuable comments that are very helpful in improving the manuscript.

**References**

Banerjee, S. and Zare, R. N.: Syntheses of isoquinoline and substituted quinolines in charged microdroplets, Angew. Chem., 127, 15008-15012, 2015.

Hecobian, A., Zhang, X., Zheng, M., Frank, N., Edgerton, E. S., and Weber, R. J.: Water-Soluble Organic Aerosol material and the light-absorption characteristics of aqueous extracts measured over the Southeastern United States, Atmos. Chem. Phys., 10, 5965-5977, 2010.

Cheng, X., Chen, Q., Li, Y., Huang, G., Liu, Y., Lu, S., Zheng, Y., Qiu, W., Lu, K., Qiu, X., Bianchi, F., Yan, C., Yuan, B., Shao, M., Wang, Z., Canagaratna, M. R., Zhu, T., Wu, Y., and Zeng, L.: Secondary Production of Gaseous Nitrated Phenols in Polluted Urban Environments, Environ. Sci. Technol., 55, 4410-4419, 10.1021/acs.est.0c07988, 2021.

Huang, R. J., Yang, L., Shen, J., Yuan, W., Gong, Y., Guo, J., Cao, W., Duan, J., Ni, H., Zhu, C., Dai, W., Li, Y., Chen, Y., Chen, Q., Wu, Y., Zhang, R., Dusek, U., O'Dowd, C., and Hoffmann, T.: Water-Insoluble Organics Dominate Brown Carbon in Wintertime Urban Aerosol of China: Chemical Characteristics and Optical Properties, Environ. Sci. Technol., 54, 7836-7847, 10.1021/acs.est.0c01149, 2020.

Lin, P., Rincon, A. G., Kalberer, M., & Yu, J. Z.: Elemental Composition of HULIS in the Pearl River Delta Region, China: Results Inferred from Positive and Negative Electrospray High Resolution Mass Spectrometric Data, Environ. Sci. Technol., 46, 7454-7462, 2012.

Lu, C., Wang, X., Li, R., Gu, R., Zhang, Y., Li, W., Gao, R., Chen, B., Xue, L., and Wang, W.: Emissions of fine particulate nitrated phenols from residential coal combustion in China, Atmos. Environ., 203, 10-17, 2019.

Lyu, R., Shi, Z., Alam, M. S., Wu, X., Liu, D., Vu, T. V., Stark, C., Fu, P., Feng, Y., and Harrison, R. M.: Insight into the composition of organic compounds ($\geq$ C 6) in PM 2.5 in wintertime in Beijing, China, Atmos. Chem. Phys., 19, 10865-10881, 2019.

Sun, Y., Tang, J., Mo, Y., Geng, X., Zhong, G., Yi, X., Yan, C., Li, J., & Zhang, G.: Polycyclic aromatic carbon: a key fraction determining the light absorption properties of methanol-soluble Brown carbon of open biomass burning aerosols,

Environ. Sci. Technol., 55(23), 15724-15733, 2021.

Wang, L., Wang, X., Gu, R., Wang, H., Yao, L., Wen, L., Zhu, F., Wang, W., Xue, L., Yang, L., Lu, K., Chen, J., Wang, T., Zhang, Y., and Wang, W.: Observations of fine particulate nitrated phenols in four sites in northern China: concentrations, source apportionment, and secondary formation, Atmos. Chem. Phys., 18, 4349-4359, 10.5194/acp-18-4349-2018, 2018.